# Art Therapy Open Studio and Teen Identity Development: Helping Adolescents Recover from Mental Health Conditions

**DOI:** 10.3390/children9071029

**Published:** 2022-07-11

**Authors:** Linda J. Kelemen, Liat Shamri-Zeevi

**Affiliations:** 1Wurzweiler School of Social Work, Yeshiva University, New York, NY 10033, USA; 2Art Therapy Department, The Academic College of the Society and Arts, Netanya 4237927, Israel; liatshamri@gmail.com

**Keywords:** OS-ID, art therapy, open studio, adolescents, identity development, supported autonomy, social anxiety

## Abstract

Adolescent identity development is driven to a significant degree by peer interaction. However, when mental health conditions (MHC) or other crises separate teens from their peers, their identity development can be slowed or arrested. We developed a unique open studio intervention (OS-ID) that could facilitate identity development in teens recovering from MHC, and incorporated this intervention into a therapeutic day school catering to our target population. We utilized qualitative case study research to explore these students’ experiences. Over the 10-month period of our intervention, we saw positive changes in the participants’ identity development. Key elements in OS-ID include the therapists’ commitment to supported autonomy; the absence of participatory demands; the emphasis on creative process over product; the use of setting and materials to promote the healing process; the facilitators’ and participants’ witnessing the process; the privatization and protection of the participants’ creations; and the ubiquitous presence of non-threatening significant others. This OS-ID modality could be an effective mechanism for assisting socially isolated teens to manage their social anxiety, develop their identity, and transition back into their peer environments.

## 1. Introduction

Adolescence is the decade-long journey that begins at puberty’s onset (around age 10) and lasts until adulthood (age 20) [1]. Throughout this decade, healthy adolescents experience dramatic physical, cognitive, and emotional changes, along with the development of more sophisticated social skills and awareness of their individual identity. Adolescents navigate through their identity development by sifting through who they do and do not want to be, and examining the values and behavior of those who positively or negatively impact their lives [2]. Mental health professionals increasingly emphasize the role played by a teen’s positive environment in encouraging emotional growth, social adjustment, self-awareness, and identity [3]. While teens’ positive and negative interactions with adults influence their identity, peer group interactions are even more impactful [4].

The United Nations Human Rights Council replaced the term “mental illness” with “mental health conditions” (MHC). This new terminology is part of a larger shift in emphasis, from pathology to recovery. It represents a transition from a medical approach to one focused on wellbeing (Tuaf & Orkibi, 2019) [5]. When mental health conditions (MHC) (or other crises (for example: war, natural disasters, pandemics, and chronic illness) separate a teen from their peer group, or when the adults and peers in their environment are not healthy themselves, then their developmental opportunities may be compromised, and the teen may miss out on significant emotional and social growth [6]. Their identity may be amorphous, conflicted, or embryonic [7]. The APA Dictionary (n.d.) defines identity as: “an individual’s sense of self defined by (a) a set of physical, psychological, and interpersonal characteristics that is not wholly shared with any other person and (b) a range of affiliations (e.g., ethnicity) and social roles.”.

Therapeutic interventions may be needed to foster these teens’ identity development. Generally, therapists facilitating teen identity development have employed exclusively verbal forms of psychotherapy. However, some teens recovering from MHC might use words to hide, and may retreat to the verbal realm to conceal conscious or subconscious psychological injuries. Additionally, some experiences and emotional states are beyond words: depression, trauma, loss, and anxiety are just a few examples of emotional states and experiences that are difficult to express for teens struggling with MHC [8].

When a teen recovering from MHC requires more than words, or something different from words, practitioners may introduce the use of semiotics and symbols as an alternative method of conceptualizing and expressing emotions and experiences. As Lakh et al. posit, “Symbols are images representing our mental content in a tangible form” [9]. This is where art therapy can sidestep therapeutic obstacles rooted in verbiage [10]. Art therapy is a form of psychotherapy that uses art media as its initial mode of communication. Art therapy circumvents the initial need for words. Image-making, and the viewing and subsequent discussion of those images, prepares participants to find words to describe their experiences [11]. However, little research has been done on the use of art therapy specifically for the development of identity in teens suffering from MHC. This is our focus.

Art therapy Open Studio (OS) is a Humanistic form of group art therapy where there is no explicit goal to accomplish, and which focuses on process over product. The OS is based on the ‘art as therapy’ approach, that emphasizes the healing quality of the creative process itself [12]. OS does not require individuals to actively interact with other group members. In OS, no project directives are given, and the model adjusts to the participants’ needs and context. Unlike some forms of group art therapy, OS allows participants to be the interpreters of their own work and the ones who determine when their process is finished [13]. Importantly, OS insists upon the individual’s focus on their own creative process within a group setting. This constitutes one of OS’ crucial priorities and strengths [14], and raises the question of whether it could be a first-tier intervention for identity development in teens suffering from MHC.

Anxiety-ridden teens may find focusing on themselves in a one-on-one therapeutic setting confrontative. They may prefer the comradery, protection, and identity that a peer group can offer [15]. However, traditional group therapy may not provide teens recovering from MHC with enough emotional safety. In contrast, OS simultaneously facilitates self-exploration and peer interaction, but without activating debilitating anxiety levels [16]. Like traditional group practitioners, OS practitioners consider how long each session should extend, and they design their respective programs focusing on clinical considerations, like group composition, the therapist/s’ role within the group, and the therapist/s’ perception of the clients.

Some of OS’s many healing elements include experiencing a variety of art materials, image-making, reparative side-by-side play, and/or mutual aid. It always includes the participants’ efforts and creation being witnessed by the therapist [17]. When therapists bear witness, they not only acknowledge the participant’s experience; they also affirm that experience’s reality, and grant it meaning and significance.

OS may help heal using another mechanism as well: The process of working with artistic materials itself exploits brain plasticity. Multiple studies demonstrate that physical actions and sensory experiences modify neural development, potentially accentuating identity development [18]. All these factors encouraged practicing OS with teens whose identity development was compromised by MHC.

## 2. Programmatic Elements of OS-ID

The Open Studio Identity Development (OS-ID) intervention we will present in this article was incorporated into a therapeutic day school catering to the unique needs of students who were recovering from MHC. (The first author served as principal facilitator of the OS. The second author served as supervisor of the OS) The school followed a ten-month academic year. Students were between 13–17 years old, came from different socio-economic backgrounds and were mostly female. In total, 13 students experienced OS-ID within this 10-month period, and eight students consistently attended OS-ID weekly for a full ten-month academic school year Table 1. This school was created for teens who had lost their academic standing, either because of hospitalization, truancy, or being homebound. Some were removed from their original schools because of psychiatric hospitalization; others did not require hospitalization, but their psychiatric issues made it impossible to thrive in a mainstream school; and still others removed themselves by refusing to leave their bedrooms for extended periods of time Table 2. Each student lost months or years of consistent school peer group interaction and identity development, making them an ideal cohort for our OS-ID intervention.

Therapy did not only take place in the OS-ID setting. The school’s goal was to prepare students recovering from MHC for transition back into community life, and so it approached every aspect of the day with a therapeutic mindset. Student activities like meal preparation, dining, kitchen clean up, animal-assisted therapy, individualized education, field trips, and community service were all choreographed to provide therapeutic benefit, just like OS-ID sessions.

We chose a qualitative research format to explore how and why OS-ID impacts its participants, rather than the frequency of specific outcomes in a client population [19]. In keeping with qualitative research principles, we focused on the students’ interactions with the setting, their peers, the therapists, and above all on their experiences with the art materials and their creations. Our case study illuminates one healing path for students recovering from MHC and subsequent delayed identity development, even though it cannot definitively establish the relative effectiveness of OS-ID versus other modalities.

Our OS-ID model is built on the artist Edward Adamson’s previous work. Adamson established the first OS inside a psychiatric hospital in the mid-20th century [20]. Since Adamson’s primary aim was to facilitate patients’ self-expression through art, he provided no guidance nor direction during the patients’ creative OS experience [21]. Additionally, Adamson did not interpret his participants’ image making. Rather, he provided a calm and patient environment with appropriate materials to help facilitate the group’s creative process [19]. This contrasts with OS facilitators who model the creative process by creating alongside the other group members [22].

The OS-ID method is open-setting, i.e., the therapist provides a setting with minimal instructions. The therapist does designate the intervention’s time and place, and chooses which materials will be available, but everything else is up to the participant. OS-ID meets minimally once a week for 1.5–3 h. As in all art therapy programs, the therapist conducts an initial, one-time session in which students co-author a contract, detailing what they need to feel safe and secure in their group setting. As the therapist solicits from the students’ phrases describing what they need to feel safe within the OS-ID group, the students write these phrases on one large piece of cardboard. Our students wrote phrases such as: “even mistakes are beautiful”, “accept me for all my colors”, and “see me without judgement”. However, once the OS-ID sessions commence, there are no ceremonies, i.e., session introductions or wrap-ups. Beyond the singular guideline that participants cannot damage themselves, others, the setting, or anyone’s art creations, there are no expectations. This idea that everything-can-be-explored links OS-ID to Adamson’s early model of OS. The goal in providing such freedom and opportunity for introspection (and resulting identity development) is to construct an accessible bridge for these teens from hospitalization to community integration.

Minimally, OS-ID requires co-therapists. This permits one therapist to supervise the session even if a student is triggered and requires special attention. The therapists set up a buffet of different art materials in the middle of a table large enough to accommodate all students and therapists. The therapists also set up additional tables and chairs around the room to accommodate those students who may choose to create alone or in pairs. The room should have a sink for water so that the students won’t hesitate to use messy materials, and so that they may remain within the liminal space for the duration of the session. The art material buffet should include the same basic art-making materials each time, such as lead and colored pencils, markers of all sized tips and colors, white glue, oil pastels, gouache and paint brushes, clay and plasticine, scissors and magazines for collage, and papers of assorted sizes and colors. This sort of reliable environment subtly cultivates the students’ sense of security, which in turn facilitates self-exploration and affirmation. Ideally, the buffet also includes specialty items, such as glitter, ingredients for slime, rocks and shells, assorted materials with needles and thread, glue guns, plaster casting strips, and wood.

We chose to secure more dangerous materials (like wood carving tools). These materials were held by one therapist and distributed only to those who would sit near this therapist. All dangerous tools had to be returned and counted before the session ended to ensure everyone’s safety. Additionally, woodworking had its own designated area in the room; so did sewing, and so did the hot glue gun.

We did not encourage students to show each other their work. Rather, we cultivated a tacit agreement, an unstated understanding, that the art creation is personal and represents some aspect of its image maker. As students got to know and trust each other, they increasingly shared their work and their reflections with each other voluntarily.

The therapists did not ask the students to help set up or clean up. As the relationships between the therapists and the students became warm, empathetic, and sincere, students spontaneously came early and stayed late to be with each other and the therapists as they set-up and cleaned-up together.

To expand opportunities for identity development, our intervention embraces a principle we dubbed ‘supported autonomy’. Making personal choices contributes to identity [23] and therefore people should be given as many choices as possible within a safe, contained environment. The OS-ID model echoes this mindset. Each student learns how to be resourceful, make positive choices, select materials, and determine how to use them, as discussed below in our case studies. They also choose to work alone or in a group. This is crucial for students with MHC struggling with identity development, as it maximizes opportunities for the students to introspect, discover, and act upon their preferences. We do not need to interrupt long moments of silence, as they may suggest that students are having an internal dialogue to contemplate and choose how they want to express themselves. Indeed, Winnicott [24] posits that one’s ability to be alone while being with another person is a significant marker of emotional maturity.

We included several structural elements to minimize the inherent power imbalance between teens and adults. For example, facilitators and students focus on learning from each other through the creative process within this transitional space [25]. The OS-ID model reduced this hierarchy even further: OS-ID facilitators and students are not equal foci of the process. The facilitators are the students’ assistants and support-staff. They function as the third hand and witness for these teens, providing guidance, emotional and creative support, empathy, and guarding the setting to ensure physical and emotional safety [17].

Some students may know what materials they want to use and projects they want to make. Others may take their time exploring materials or watching what others create. Some may want facilitator interaction or suggestions, while others may not. No one is required to interact with anyone else, but anyone could. If a student leaves before the end of the session, one of the therapists should gently encourage the student to return.

Developing identity requires taking the (often courageous) step of affirming our choices. OS-ID therapists support this emotional process by protecting the privacy of the choices made by each student: The OS-ID is the students’ time to speak without words, to open their wounds to themselves and, only if they wish, to each other or to the facilitators. OS-ID therapists handle the students’ creations as one would handle a therapeutic dialogue: respectfully, with consideration and confidentiality. Therefore, after every session, the students’ creations are carefully packed away by the therapists. The OS-ID students’ silent dialogues between their art creations and themselves remained private within this therapeutic environment, as did their tumult, quiet times, jokes, rollicking and laughter. By making it safe for these students recovering from MHC to be themselves in OS-ID sessions, both in their art and in-session conduct, OS-ID facilitates identity development.

The therapist’s role in OS-ID is not obvious to the untrained eye. OS-ID is not an art class, and the therapist is not an art teacher. OS-ID therapists seek to establish a safe, calm, and patient environment by approaching their clients with warmth, acceptance, and empathy. They guide their clients in establishing a contract that will enhance their feelings of security and safety. They encourage supported autonomy, and they guard their clients’ moments of silent contemplation. They help clients appreciate that their art creation is personal and need not be shared with anyone. It is a therapeutic dialogue between the client and their art creation, and therefore deserves confidentiality and preservation. The therapist attends to the process of preserving the art creations in a safe and private space until the clients’ departure from the school or treatment facility. If a client is triggered or expresses a desire to prematurely end an OS-ID session, their therapists provide emotional support and help the client return to their creative explorations. OS-ID therapists fulfill this role to help their clients discover and develop their own identities.

## 3. Case Studies

The two case studies below illustrate the OS-ID model, its potential strengths, and its applications. Student names and non-essential background details have been changed.

### 3.1. Ethan

Ethan is a tall, physically fit 16-year-old who was raised in a small, urban apartment with his mother and father, aunt, and two sisters. Like his mother, Ethan suffers from depression, anxiety, and subsequent low self-esteem. Further, Ethan’s masculine identity development lagged. At age 12, he claimed that male peer interactions brought on his anxiety attacks. According to Ethan, he first attended school only sporadically and eventually dropped out to avoid same-sex social interactions. He retreated to his bedroom for the next two years. He had psychiatric hospitalization twice during this two-year period due to suicide ideation. His psychiatrist prescribed anti-depressant and anti-anxiety medications to help with his daily functioning and to ameliorate his emotional instability. At age 16, Ethan transferred from the hospital’s outpatient day school to our school.

Ethan spent his first OS-ID session standing silently at the studio’s doorway while wearing his backpack and clenching his fists. He asked me if he could bring his guitar for the next session, explaining that music is artistic too and that he didn’t like working with art materials. His request felt to me like a defense mechanism, a way to remain safely distant from engagement. I encouraged Ethan to bring his guitar to the next session. When Ethan arrived for session two, he chose to place his guitar in the corner of the room and sat with the group. He still hesitated to engage in art creation himself, but spent the session observing his peers interact with their chosen materials.

By the third session, Ethan chose to work with soft materials, such as pipe cleaners and yarn, braiding colorful bracelets and tiaras as gifts for the female students. Later, he confided that, even though he enjoyed working with soft materials, he wanted to experience the powerful feeling of carving wood. We discussed this option and decided to bring wood and carpentry tools into the OS-ID. Students who wanted to carve wood using these tools needed to sit near the therapist who was responsible for guarding and distributing carving tools. For the next several sessions, Ethan sat with this therapist and carved his name and other symbols into wood. More students joined them, including some male students, and Ethan seemed comfortable with them. As Ethan started exploring different materials, such as dripping wax from lit candles onto large pieces of translucent Bristol board or encapsulating flowers into hot glue, we noticed his social behavior begin to change. Despite his previous hesitance to socially interact with male peers, he confided to me that now he “hangs out with the boys,” and he stopped creating gifts for the female students. Ethan seemed to be successfully challenging his own anxiety over interacting with males.

One of Ethan’s last creations was dripping wax across a 23 × 33 cm Bristol board (Figure 1). When Ethan said he was finished, I asked him if we could lift the board up towards the light. He agreed. Ethan and I gazed at his art creation together. I reflected that this technique reminded me of stained glass. One of the male students noticed us poring over Ethan’ creation and blurted out, “It looks like sperm!” Ethan turned to him in shock. There was a moment of nervous contemplation, and then he muttered, “Oh, uh huh,” shrugged his shoulders with resignation, and gave everyone a look of acquiescence. It was as if he had been accused of displaying masculinity and he tentatively concluded that doing so would be okay. But then he may have had second thoughts. Ethan delayed leaving the OS-ID until his friends were outside. Then Ethan put his art creation down on the table, said “throw it out,” and headed toward the door. I gently reminded him, “We keep everything we create”. He stopped and turned towards me, looking back and forth between me and his art creation. I reassured him, “At the last session, we can sort through what you want to keep, what you want to discard, and how you want to discard it”. “I know, but why?” he inquired. “Everything you create is a part of you,” I explained, “and we accept all of you here.” Ethan seemed to settle back into comfort with his masculinity. He walked back to the table, picked up his art creation, and handed it to me. “Thank you”, he said, and then he joined his friends outside the OS-ID Table 3.

This was a significant, but small step on Ethan’s road to recovery. At the end of our OS-ID intervention, Ethan’s psychiatric profile remained complex. Nonetheless, OS-ID helped Ethan ease back into the world of male companionship.

### 3.2. Liora

Liora is a 15-year-old girl who lives with her mother, father, and four younger siblings in a large house at the edge of a farming community. When Liora was seven years old, her mother was diagnosed with borderline personality disorder (BPD). Her mother’s struggles with BPD included periods of self-isolation, verbally abusive outbursts, and suicide attempts that were followed by lock-down hospitalization. Even from a young age, Liora served as a parental substitute, feeding and bathing her siblings, cleaning the house, cooking, washing the laundry and dishes, and organizing her mother’s medical appointments and medications.

Liora was unique in our population insofar as she excelled at peer relationships. She was not an extrovert, but exuded social confidence and friendliness, despite her low self-esteem. She maintained many peer friendships and enjoyed group activities. Her challenge in developing identity was her parents’ near-total absence. Her mother’s psychiatric disorder, and her father’s frenzied efforts to manage that crisis, left her without enough significant physical or emotional support at home. That, combined with her own learning disabilities, cultivated anxiety. Because she couldn’t manage mainstream school academically and emotionally, Liora dropped out at age 15 and enrolled in our therapeutic day school. She not only needed a therapeutic and academically personalized environment to help her cope; she needed to develop her own identity by relating to reliable adult role models.

Liora was artistic and appreciated the opportunity to express herself using different materials. In her first OS-ID session, she chose a graphite pencil to draw a 18 × 24 cm picture of herself tied up and alone (Figure 2). People in pursuit of control gravitate toward graphite pencils, and it isn’t hard to imagine why someone raised without enough parental protection and guidance might intuitively try to control her environment. The theme of being bound and alone appeared in many of her subsequent art creations. For example, she constructed a jail cell using plaster casting and paper mâché, and a broken heart using colored paper and wood. These themes are consistent with a child’s experience of having to run a household on her own. During her OS-ID process, the theme of her art creations transitioned away from entrapment, and she started using paint to illustrate homes. Additionally, while her early art creations consisted only of images, her later creations segued toward the verbal, including words from popular poems and lyrics. It was as if she was discovering, for the first time, the words to express her trauma.

Not surprisingly, Liora initially showed no interest in the two therapists. Without reliable parents in her life, Liora learned not to expect emotional support from adults. Her teachers taught her. Her school principal disciplined her. Her psychiatrist medicated her. But adults didn’t empathize or befriend her, and to her that seemed normal. She saw no reason to bond with us. Unlike other students who would even arrive early or stay later to talk with the therapists, Liora rarely spoke with or established eye contact with the therapists even during the OS-ID.

However, after months of OS-ID sessions Liora turned a corner. She picked up her painting materials and started to walk out of the OS-ID room, calling to me, “I am going to paint outside”. OS-ID encourages students to remain part of the group, so I responded, “Can you paint outside, but on the porch, so you can still be with me?”. Liora paused, looked at me directly for the first time, smiled warmly and said, “I’ll paint with you here, inside”. She perched herself on the windowsill with her art materials and painted an 18 × 24 cm picture of the houses she could see ensconced in the hillside (Figure 3). Then, as if to reiterate to us her interest in the relationship, Liora stayed after the OS-ID session. She helped me clean up and spoke to me about her ideas for her next painting (Table 4). Perhaps her witnessing other students increasingly trusting us built Liora’s own sense of safety and trust. In this way, the OS-ID provided Liora a safe space to decide when and how to approach us. She used art to connect to us. Her anxiety, at least with her OS-ID therapists, was ameliorated.

## 4. Discussion

Social anxiety is common among teens with MHC, especially those who have been isolated from their peers for an extended period [26]. The OS-ID model attempts to ameliorate this and other social anxieties by minimizing social demands. Participants remain part of the group, even if they resist being emotionally open and decline to share their thoughts, feelings, and art creations. Social interaction and participation become a choice, not an unwritten obligation. Teens can choose to create relationships with others through their mutual interactions with the materials and the image-making process, without the peer pressure to emotionally engage [27].

The OS-ID model also attempts to ameliorate anxiety with its unique approach to “significant others”. A “significant other” is an individual, specifically a peer or parental figure, who profoundly influences another person’s identity and socialization [27]. Significant others can enhance someone’s sense of self and reduce anxiety by providing mutual validation. However, for teens struggling with MHC, a significant other can create spoken and unspoken obligations and precipitate anxiety [28]. The OS-ID model presents this vulnerable population with non-threatening significant others: The OS-ID therapists constitute adult significant others, and the other students become the peer significant others. Both provide mutual validation, but without any associated social demands.

Reducing anxiety within the OS-ID is a means to an end, that end being identity development. Our program encourages students to explore and affirm their identity. Within an environment of supported autonomy, OS-ID invites students to ask themselves what they would like to create, and how they would like to create it. Students discover the answers to these questions by contemplating what they want and therefore who they are. Artistic freedom thus primes the process of self-discovery and therefore encourages identity development.

In OS-ID, materials and art creations are not always the active ingredients stimulating identity development. Sometimes, just the social setting is reparative. Ethan had an attenuated masculine identity, which he attributed to having neither male friendships nor a bond with a familial adult male. He shied away from bonding with adult males, and explained that this was out of concern that they would mock him for being effeminate. Despite gravitating toward females, Ethan expressed a desire to establish male friendships. In OS-ID, Ethan discovered he could find acceptance and validation from male peers. His integration into a male peer group was his choice and proceeded under his control and at his pace. Ethan saw aspects of his own personality in the male therapist, and therefore he felt that the male therapist truly understood him [29]. This twinship helped Ethan feel valued and boosted his self-esteem. Ethan’s burgeoning self-esteem reduced his social anxiety and stimulated his identity development.

In other cases, materials and art creation are more central to and illustrative of a student’s reparative process. When Liora first entered the OS-ID, she created entrapment symbols (chains, handcuffs, prison, etc.), and she did so using resistive materials. The contrast between her early art creations and her later creations suggested internal changes: She segued to painterly depictions of houses, using fluid media. According to Hinz’ Expressive Therapies Continuum, using resistive materials is associated with more cognitive experiences, while using fluid materials is associated with more affective experiences [30]. According to this theory, Liora’s transition from resistive to fluid materials implies a modal transition from cognitive to affective experience. This could be significant if emotional involvement is prerequisite to identity development.

After prisons became homes, her art creations changed again: Liora began incorporating words into her images, inserting poetry and lyrics that conveyed hope, passion, and belonging. She transitioned from expressing herself exclusively with images to expressing herself with images and others’ words. This may have been a precursor for her eventually describing her feelings using her own words.

Even in Liora’s case, however, the OS-ID social setting was also a reparative element. Liora arrived at our school lacking trust and interest in relating to adults. Her BPD mother distanced her with caustic insults, and her father was too overwhelmed caring for his wife to give Liora adequate encouragement and affection. Because Liora was both her siblings’ and her own primary caretaker, she grew up in a home that, from her perspective, lacked parents. Therefore, adults had no place in her familial model. To develop meaningful relationships with adults, Liora needed to experience consistently safe and supportive interactions with adults.

During the span of this OS-ID program, we saw two types of change in our students: They experienced varying levels of recovery from psychological wounds, as suggested by changes in their art creations and behavior. For example, a student who feared male-bonding developed the self-confidence to join a male peer-group; and a student who initially viewed herself as a prisoner found freedom. Second, their social anxiety lessened, which allowed them to interact with significant others while exploring their creative process.

Although Ethan and Liora had different developmental challenges, OS-ID proved sufficiently broad and flexible to simultaneously serve both of their therapeutic needs.

### Limitations

The literature describes the ideal OS session as being longer than the common 50-min clinical session. This allows the artmaking experience to progress naturally toward conclusion, as opposed to being artificially truncated by scheduling limitations. It also permits the participant to engage in the creative process more fully. However, this school could not accommodate OS-ID sessions longer than 1.5 h. A program providing a longer OS-ID session might produce experiences and outcomes radically different from those we observed. This calls for an in situ longitudinal study. Additionally, the OS-ID program only convened once a week. It is possible that students would have been more impacted by meeting more often.

Because ethical concerns prevented us from including a control group in our study, we cannot distinguish the impact of our intervention from the impact of the client-therapist relationship, the school’s larger therapeutic environment, influences from home, and the potentially positive impact of the passage of time. Future studies could focus on developing ethical approaches to contrasting the impact of OS-ID with the impact of these other factors.

## 5. Conclusions

To summarize, there were several key elements in Ethan’s and Liora’s OS-ID healing experience: their freedom to contribute to the therapeutic contract; the therapists’ commitment to supported autonomy; the absence of participatory demands; the emphasis on creative process over product; the use of setting and materials to promote the healing process; the facilitators’ and participants’ witnessing of the process; the privatization and protection of their creations; and the ubiquitous presence of non-threatening significant others Table 5.

During the span of this OS-ID program, we saw two types of change in our students: Individually, they experienced varying levels of recovery from psychological wounds. For example, a student who feared male-bonding developed the self-confidence to join a male peer-group; and a student who initially viewed herself as a prisoner found freedom. Their social anxiety lessened, which allowed them to interact with significant others while exploring their creative process. We saw evidence of these individual and social changes in both their art creations and their behavior.

The OS-ID model described here constitutes a tool worthy of consideration for helping teens who have been separated from their peers develop their identities and socially stabilize. We are writing in the shadow of the COVID-19 pandemic, and one can only wonder what the effects will be of isolating millions of teens from their peers during lockdowns and school shutdowns [6]. Educators and therapists may seek interventions to assist these youth in managing their social anxiety, developing their identity, and transitioning back into their peer environments. This OS-ID modality could be an effective mechanism for this and similar crises.

## Figures and Tables

**Figure 1 children-09-01029-f001:**
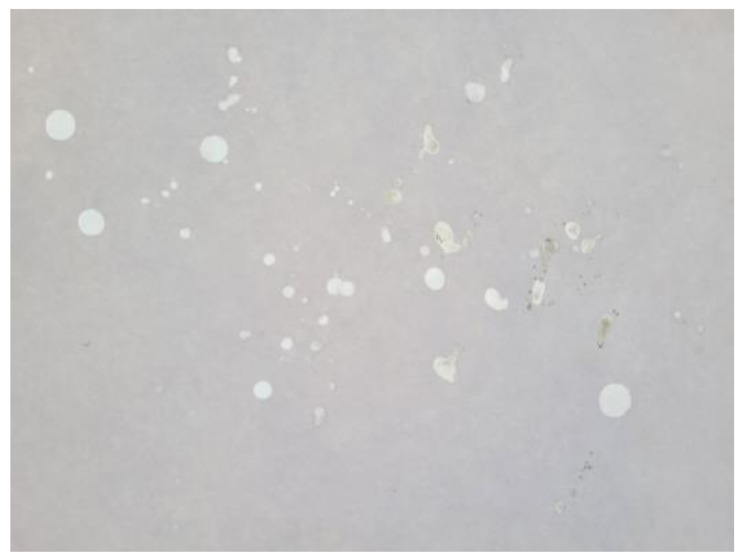
Ethan’s wax drippings on Bristol board.

**Figure 2 children-09-01029-f002:**
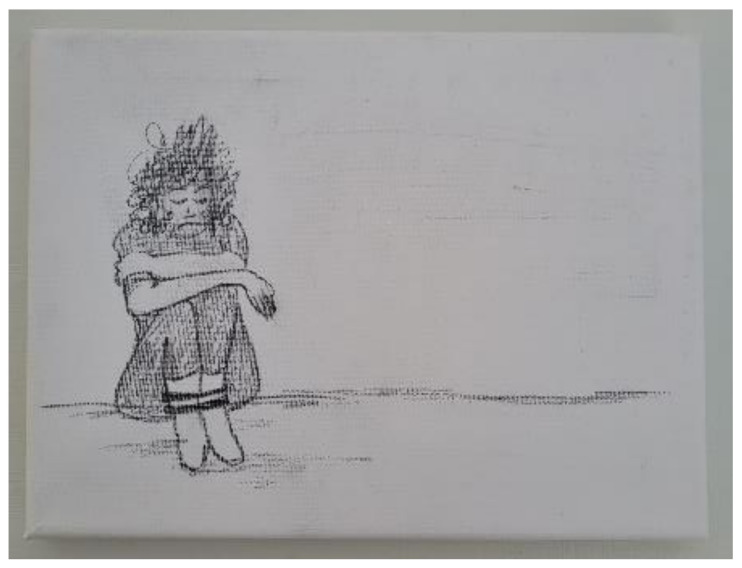
Leora’s drawing of a girl bound and alone.

**Figure 3 children-09-01029-f003:**
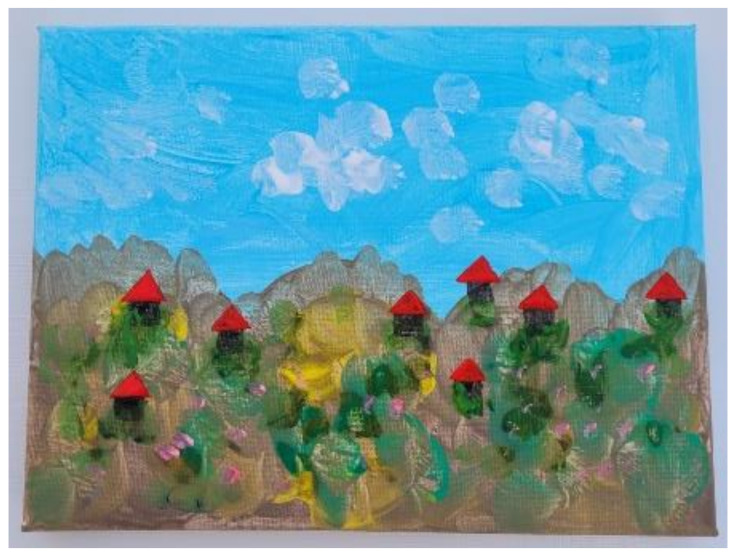
Liora’s painting of houses in the hills.

**Table 1 children-09-01029-t001:** Case Study Demographics.

Category	Number
*n* of schools or regions targeted	(*n* = 1)
*n* of total students participating in OS-ID within a 10-month period	(*n =* 13)
*n* of students participating in OS-ID for the full 10-month academic school year	(*n* = 8)
Time scale of students’ lost academic standing	3 months to 4 years
Reason for previous lost academic standing: Homebound	(*n* = 4)
Reason for previous lost academic standing: Hospitalization	(*n* = 6)
Reason for previous lost academic standing: Truancy	(*n* = 3)

**Table 2 children-09-01029-t002:** Student Demographics.

Student ID	Gender	Age	Months Attending OS-ID	Reason for Previous Lost Academic Standing
S1	Female	13	10	Homebound
S2	Female	14	3	Hospitalization
S3	Female	14	10	Truancy
S4	Female	15	10	Truancy
S5	Female	16	2	Hospitalization
S6	Female	16	10	Hospitalization
S7	Female	17	10	Homebound
S8	Female	17	10	Hospitalization
S9	Female	17	2	Homebound
S10	Male	14	10	Hospitalization
S11	Male	16	1	Truancy
S12	Male	16	10	Hospitalization
S13	Male	17	8	Homebound

**Table 3 children-09-01029-t003:** Summary of Ethan’s ten-month OS-ID process.

Session	Event Summary
Session 1	Ethan stood silently at the studio’s doorway while wearing his backpack and clenching his fists, did not participate with the group, and asked to bring his guitar to the next session.
Session 2	Ethan brings his guitar, but places it in the corner of the room and sits with the group. He spends the session observing his peers interacting with their chosen art materials.
Session 3	Ethan working with pipe cleaners, a soft material, and creates gifts for the female students. He requests from the therapists to work with wood, a harder material.
Session 4	Ethan sits next to the male therapist and starts to explore wood carving. Other students join them, including two male students.
Next several sessions	Ethan continues to carve wood alongside the male therapist and other students. Ethan explores other materials too, such as wax dripping, and hot glue. His social behavior begins to change. He stops creating gifts for the female students and prefers to socialize with the male students.
One month before ending	One of Ethan’s dripping-wax creations is challenged by one of the male students, noticing that Ethan’s wax drippings look like sperm. Initially, Ethan seems threatened by this confrontation, but is comforted by the female therapist reminding him that all parts of him are accepted here in OS-ID.

**Table 4 children-09-01029-t004:** Summary of Liora’s ten-month OS-ID process.

Session	Event Summary
Session 1	Liora chooses a graphite pencil, an easily controlled art material, to draw a picture of herself tied up and alone.
Next several sessions	Liora constructs a jail cell using plaster casting and paper mâché, and a broken heart using colored paper and wood.
After four months	The theme of Liora’s artwork transitions away from entrapment, and she starts using paint to illustrate homes (a more fluid material).
After seven months	Liora includes poetry and lyrics from popular poems and lyrics. Liora begins to show interest in socializing with the therapists.
After eight months	Liora picks up her painting materials and states that she is going to paint outside. Instead, she chooses to stay with the group and paints inside. She makes eye contact with the therapists and speaks directly to them. Liora stays after the OS-ID session, helps clean up, and discusses her ideas for her next painting with the therapists.

**Table 5 children-09-01029-t005:** Eight key elements in Ethan’s and Liora’s OS-ID healing experience.

Key OS-ID Elements	
1.	Freedom to contribute to the therapeutic contract
2.	The therapists’ commitment to supported autonomy
3.	The absence of participatory demands
4.	The emphasis on creative process over product
5.	The use of setting and materials to promote the healing process
6.	The facilitators’ and participants’ witnessing of the process
7.	The privatization and protection of their creations
8.	The ubiquitous presence of non-threatening, significant others

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
