# Peer review of "Art Therapy Open Studio and Teen Identity Development: Helping Adolescents Recover from Mental Health Conditions"

_children, 2022, doi:10.3390/children9071029_

Round 1
Reviewer 1 Report
This is the report of a community case study in which two cases are being described, both of young adults. After a 10-month observation period both adults showed improvement in their identity development. What exactly this can be attributed to (the art therapy / open studio intervention) or the regular attention by the therapist cannot be distinguished. The role of the therapist and their influence on the client`s positive development could be discussed further.
Author Response
Thank you for your insightful recommendations. We have now included the following paragraph under “Limitations”:
Because ethical concerns prevented us from including a control group in our study, we cannot distinguish the impact of our intervention from the impact of the client-therapist relationship, the school’s larger therapeutic environment, influences from home, and even just the potentially positive impact of the passage of time. Future studies could focus on developing ethical approaches to contrasting the impact of OS-ID with the impact of these other factors.
Additionally, we added this paragraph detailing the therapist’s role:
The therapist’s role in OS-ID is not obvious to the untrained eye. OS-ID is not an art class, and the therapist is not an art teacher. OS-ID therapists seek to establish a safe, calm, and patient environment by approaching their clients with warmth, acceptance, and empathy. They guide their clients in establishing a contract that will enhance their feelings of security and safety. They encourage supported autonomy, and they guard their clients’ moments of silent contemplation. They help clients appreciate that their artwork is personal and need not be shared with anyone. It is a therapeutic dialogue between the client and their art creation, and therefore deserves confidentiality and preservation. The therapist attends to the process of preserving artwork in a safe and private space until the clients’ departure from the school or treatment facility. If a client is triggered or expresses a desire to prematurely end an OS-ID session, the therapists provide emotional support and help the client return to their creative explorations. OS-ID therapists fulfill this role to help their clients discover and develop their own identity.
These were crucial improvements, and we are thankful for your help.
Reviewer 2 Report
The purpose of this study was to confirm the effectiveness of the unique open studio intervention (OS-ID). Developing an approach to recovery for teens in various environments and crises is very important, and this study is invaluable in examining directions.
Major Comments
This study takes the form of a narrative approach and presents two case studies. With respect to the case studies, the authors are carefully described and entirely understandable.
The two cases occupy an important place in this research paper. If possible, it would be helpful to add explanations of the two cases in a tabular format, in addition to the current presentation format, to further the understanding of its contents. I believe the author's intentions would be conveyed more clearly by summarizing the information in a concise tabular form.
Regarding the open studio intervention (OS-ID) that was the subject of this study, there is a lack of information on the actual implementation of the OS-ID. For example, information on when the program started, the total number of participants, the period covered, and the number of schools or regions targeted should be provided. The presentation of numerical figures will ensure the specificity of the study and make it more accurate and reliable.
In the conclusion section of this study, the authors report that they observed two types of changes in the target students. I believe these items are of great importance and should be addressed in the discussion part, if possible, to develop the discussion further. Please consider this.
Author Response
Thank you for your insightful recommendations. We have now included the following tables:
Table 1.1 Student Demographics
|
|
|
|
Table 1.2
n of schools or regions targeted |
(n=1) |
n of total students participating in OS-ID within a 10-month period |
(n=13) |
n of students participating in OS-ID for the full 10-month academic school year |
(n=8) |
Time scale of students’ lost academic standing |
3 months to 4 years |
Reason for previous lost academic standing: Homebound |
(n=4) |
Reason for previous lost academic standing: Hospitalization |
(n=6) |
Reason for previous lost academic standing: Truancy |
(n=3) |
Table 2
Nine key elements in Ethan’s and Liora’s OS-ID healing experience
1. |
Freedom to contribute to the therapeutic contract
|
2. |
The therapists’ commitment to supported autonomy
|
3. |
The absence of participatory demands
|
4. |
The emphasis on creative process over product
|
5. |
The use of setting and materials to promote the healing process
|
6. |
The facilitators’ and participants’ witnessing of the process
|
7. |
The privatization and protection of their creations
|
8. |
The ubiquitous presence of non-threatening, significant others
|
Table 3
Summary of Ethan’s ten-month OS-ID process
Session 1 |
Ethan stood silently at the studio’s doorway while wearing his backpack and clenching his fists, did not participate with the group, and asked to bring his guitar to the next session. |
Session 2 |
Ethan brings his guitar, but places it in the corner of the room and sits with the group. He spends the session observing his peers interacting with their chosen art materials. |
Session 3 |
Ethan working with pipe cleaners, a soft material, and creates gifts for the female students. He requests from the therapists to work with wood, a harder material. |
Session 4 |
Ethan sits next to the male therapist and starts to explore wood carving. Other students join them, including two male students. |
Next several sessions |
Ethan continues to carve wood alongside the male therapist and other students. Ethan explores other materials too, such as wax dripping, and hot glue. His social behavior begins to change. He stops creating gifts for the female students and prefers to socialize with the male students. |
One month before ending |
One of Ethan’s dripping-wax creations is challenged by one of the male students, noticing that Ethan’s wax drippings look like sperm. Initially, Ethan seems threatened by this confrontation, but is comforted by the female therapist reminding him that all parts of him are accepted here in OS-ID. |
Table 4
Summary of Liora’s ten-month OS-ID process
Session 1 |
Liora chooses a graphite pencil, an easily controlled art material, to draw a picture of herself tied up and alone. |
Next several sessions |
Liora constructs a jail cell using plaster casting and paper mâché, and a broken heart using colored paper and wood. |
After four months |
The theme of Liora’s artwork transitions away from entrapment, and she starts using paint to illustrate homes (a more fluid material). |
After seven months |
Liora includes poetry and lyrics from popular poems and lyrics. Liora begins to show interest in socializing with the therapists. |
After eight months |
Liora picks up her painting materials and states that she is going to paint outside. Instead, she chooses to stay with the group and paints inside. She makes eye contact with the therapists and speaks directly to them. Liora stays after the OS-ID session, helps clean up, and discusses her ideas for her next painting with the therapists. |
Additionally, information on when the program started, the total number of participants, the period covered, and the number of schools or regions targeted should be provided is now provided not only in these tables, but in the text itself.
We also added this observation to the Discussion section:
During the span of this OS-ID program, we saw two types of change in our students: They experienced varying levels of recovery from psychological wounds. Their art creations and behavior developed. For example, a student who feared male-bonding developed the self-confidence to join a male peer-group; and a student who initially viewed herself as a prisoner found freedom. Their social anxiety lessened, which allowed them to interact with significant others while exploring their creative process.
These were crucial improvements, and we are thankful for your help.